# The psychosocial burden of cutaneous leishmaniasis in rural Sri Lanka: A multi-method qualitative study

Hasara Nuwangi[1], Lisa Dikomitis[2], Kosala Gayan Weerakoon[3], Suneth Buddhika Agampodi[4,5]*, Thilini Chanchala Agampodi[1]

**1** Department of Community Medicine, Faculty of Medicine and Allied Sciences, Rajarata University of Sri Lanka, **2** Centre for Health Services Studies and Kent and Medway Medical School, University of Kent, **3** Department of Parasitology, Faculty of Medicine and Allied Sciences, Rajarata University of Sri Lanka, **4** International Vaccine Institute, Seoul, South Korea, **5** Department of Internal Medicine, Section of Infectious Diseases, Yale University School of Medicine, New Haven, Connecticut, United States of America

\* suneth.agampodi@yale.edu

**Data Availability Statement:** All relevant data are within the manuscript and its supporting information files.

## Abstract

Leishmaniasis is a tropical infectious disease affecting some of the world's most economically disadvantaged and resource-poor regions. Cutaneous leishmaniasis (CL) is the most common out of the three clinical types of Leishmaniasis. Since 1904 this disease has been endemic in Sri Lanka. CL is considered a disfiguring stigmatising disease with a higher psychosocial burden. However, there needs to be a more in-depth, holistic understanding of the psychosocial burden of this disease, both locally and internationally. An in-depth understanding of the disease burden beyond morbidity and mortality is required to provide people-centred care. We explored the psychosocial burden of CL in rural Sri Lanka using a complex multimethod qualitative approach with community engagement and involvement. Data collection included participant observation, an auto-ethnographic diary study by community researchers with post-diary interviews, and a Participant Experience Reflection Journal (PERJ) study with post-PERJ interviews with community members with CL. The thematic analysis revealed three major burden-related themes on perceptions and reflections on the disease: wound, treatment, and illness-experience related burden. Fear, disgust, body image concerns, and being subjected to negative societal reactions were wound-related. Treatment interfering with day-to-day life, pain, the time-consuming nature of the treatment, problems due to the ineffectiveness of the treatment, and the burden of attending a government hospital clinic were the treatment-related burdens. Anxiety/worry due to wrongly perceived disease severity and negative emotions due to the nature of the disease made the illness experience more burdensome. Addressing the multifaceted psychosocial burden is paramount to ensure healthcare seeking, treatment compliance, and disease control and prevention. We propose a people-centred healthcare model to understand the contextual nature of the disease and improve patient outcomes.

**Funding:** This research was carried out as part of the program ECLIPSE funded by the National Institute for Health and Care Research (NIHR - https://www.nihr.ac.uk) (NIHR200135) using UK aid from the UK Government to support global health research (H.N., L.D, K.G.W., S.B.A and T.C. A). The funders had no role in the study design, data collection, and analysis, decision to publish, or preparation of the manuscript. The views expressed in this article are those of the authors and not necessarily those of the NIHR or the UK Department of Health and Social Care.

**Competing interests:** The authors have declared that no competing interests exist.

## Author summary

In order to enhance the provision of healthcare for individuals afflicted with CL in rural Sri Lanka, it is imperative to delve into the consequences of the disease beyond its physical manifestations. Our research methodology encompassed a diverse array of approaches, including participant observation, diaries maintained by community researchers with subsequent interviews, and a Participant Experience Reflection Journal (PERJ) to gain insights into the experiences of community members affected by CL. After conducting thematic analysis of all the data sets, we identified three categories of burden that were interconnected with individuals' perceptions and encounters with the disease: Wound-related burden, Treatment-related burden and Illness-experience-related burden. People with CL were found to experience fear, disgust, and concerns related to body image stemming from the visible wounds and faced adverse societal reactions due to their disease. The treatment regimen for CL posed various challenges for patients, such as disruption of their daily lives and physical discomfort. The experience of illness became more burdensome due to anxiety/worry concerning the illness's severity, coupled with negative emotions linked to the disease. Prioritising the psychosocial burden associated with CL is essential for healthcare seeking, compliance, and disease control. We propose a context-specific, people-centred model to improve patient outcomes.

## Introduction

Leishmaniasis affects around seven hundred thousand to one million people every year [1]. It is caused by a protozoan parasite that belongs to several species of the genus *Leishmania*. This parasite is transmitted to humans by the bite of female sandflies (genera *Phlebotomus* and *Lutzomyia*). The disease is endemic in many countries across the globe, including Europe, the Middle East, Africa, and Asia [2]. Of the three main clinical types of leishmaniasis; cutaneous leishmaniasis (CL), mucocutaneous leishmaniasis (MCL) and visceral leishmaniasis (VL) [3,4], CL is the most common type which causes skin ulcers, papules, nodules, or plaques. CL is considered a disfiguring disease [5,6] which may need long-term treatment affecting the psychosocial well-being of a person.

The CL lesions are typically found in uncovered body parts, such as the face, arms, legs and neck. These can appear as one or more lesions [5]. These ulcers could be self-healing, while some can end up in scarring. The clinical manifestations of CL differ according to the species of *leishmania*, and up to 10% of cases could lead to severe manifestations such as diffuse cutaneous leishmaniasis, leishmaniasis recidivans and mucocutaneous leishmaniasis [4,7]. The nature of the psychosocial impact depends on various factors, such as the site, severity, and visibility of the lesions [8]. The disfiguration of CL often leads to stigmatisation and social exclusion, mental health comorbidities, impact on an individual's socioeconomic status and quality of life [9,10]. The global mean age-standardized burden of CL is reported as 0·58 DALYs (95% CI 0·26–1·12) per 100000 people in 2013 [11]. However, in this calculation, the researchers only considered disfigurement to estimate the years lived with disfigurement leaving out other complex psychosocial burdens attached to this disease, such as social stigma and financial and emotional burdens [12]. There is a crucial need for in-depth inquiry on individuals and communities to understand the socio-economic and psychological impact of CL in order to have better estimates for its burden [7].

Previous research has shown that the psychosocial burden of leishmaniasis significantly differs across many socio-demographic factors, populations, and cultures [10]. A study

conducted among high school students in Morocco revealed that they perceive CL as a serious disease and have concerns about their body image [13]. Similarly, in Tunisia, there have been reports of dissatisfaction with self-image and an anticipation of rejection [14]. In Sri Lanka, leishmaniasis has been identified as a disease with public health importance since 1904, the cases were reported in The Annual Administration Reports of Ceylon/Sri Lanka from 1895 to 1970 and the Ceylon Blue Book from 1821 to 1937 [15]. In 2008, leishmaniasis was re-included in the public health surveillance system [16]. Since then, a clear upward trend of reported cases has been observed [17]; in 2022 alone, more than 3000 cases were reported in Sri Lanka [18]. Despite having a recorded history of leishmaniasis of more than a century, the disease's psychosocial burden has yet to be explored in Sri Lanka.

Even though a small body of literature on the psychosocial burden attached to CL exists [10], a comprehensive qualitative exploration of the psychosocial burden attached to CL is lacking in both global and local literature. The limited global literature is inadequate to understand the context-specific issues related to this topic. This study aims to investigate the psychosocial burden attached to CL using extensive community engagement and involvement (CEI). The study will develop the evidence urgently needed to inform health policy and interventions to address the psychosocial impact associated with CL.

## Method

### Ethics statement

The ECLIPSE study was granted ethical clearance by the Ethics Review Committee of the Faculty of Medicine & Allied Sciences, Rajarata University of Sri Lanka, Saliyapura, Sri Lanka (ERC/2020/74) and Faculty of Medicine and Health Sciences, Keele University, United Kingdom (MH-200123) [19]. We will not use the names of the localities we conducted fieldwork according to the ethical principles followed throughout this project. All identifiable details of individual study participants (including names, contact numbers, and residence places) were removed or pseudonymised throughout the data sets. All study participants were assigned a unique Participant ID. Informed written consent was obtained from all study participants before participation in the study [20].

### Study design

This study was conducted as a part of a larger project: Empowering people with Cutaneous Leishmaniasis Intervention Programme to improve the patient journey and reduce Stigma via Community Education (ECLIPSE). This programme aims to understand community needs, empower communities and formulate effective policies to improve the patient journey and reduce the burden of CL, using community engagement and involvement (CEI) as a strategy in Brazil, Ethiopia and Sri Lanka [19]. We describe in this paper the qualitative design used to explore the psychosocial burden of CL in rural Sri Lanka. We used a multi-method qualitative design with two main study components; an ethnographic study on the communities (to explore the culture, context and health-related behaviours) and a qualitative study on people with CL in the selected communities (to understand the psychosocial burden encountered by people with CL within the previously explored context) (Fig 1). Hence the study design included iterative data analysis to support knowledge generation of later study components. As we were unable to continue some of the planned ECLIPSE methods (participant observation) due to the COVID-19 pandemic situation in Sri Lanka, we adapted the methods as necessary (described below) to continue knowledge generation during the pandemic [21]. The ethnographic study consisted of qualitative methods; participant observation, autoethnographic diary study and post-diary interviews. The subsequent study on people with CL

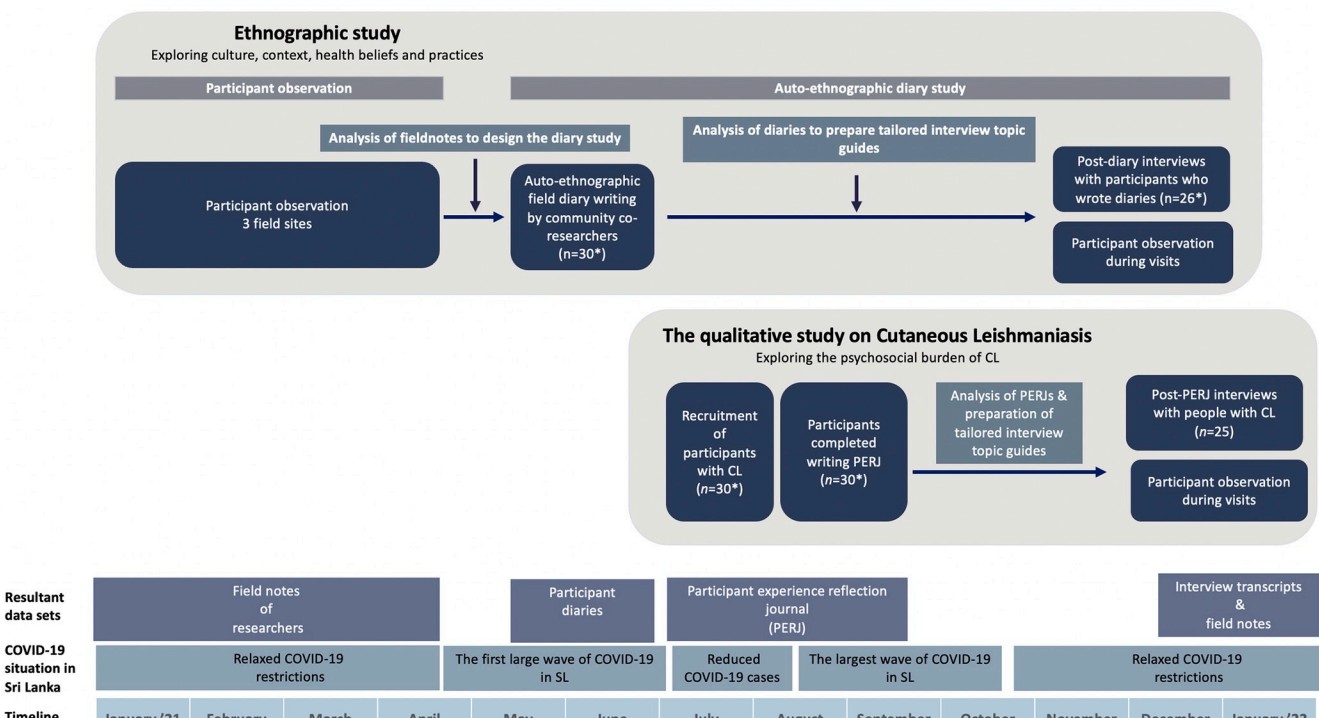

**Fig 1. Study design.**

included; Participant Experience Reflection Journal (PERJ) and combined post-PERJ interviews and participant observations (Fig 1).

## Study setting

We conducted the study in the Anuradhapura district (with over 94% rurality), with a population of 860,575 people. The majority ethnicity is Sinhalese (91%), and 90% of the population are Buddhists. The gender division of those in active employment (56.1%) is 30–35% among women. The literacy rate is at 90.5% [22]. Anuradhapura district has one of the highest CL prevelence in Sri Lanka [23]. This district consists of twenty-two administrative units called Divisional Secretariat (DS) Divisions and 694 Grama Niladhari (GN) Divisions which fall under DS divisions (Fig 2). We first selected three DS divisions with the highest prevalence of CL, based on data from Anuradhapura's Regional Director of Health Services (RDHS) office: Padaviya, Thalawa, and Nachchaduwa. We selected the three GN divisions with the highest CL prevalence at the time when the study commenced, within these three larger DS divisions.

## Component 1: Ethnographic study

The aim of the ethnographic study was to explore the cultures, community contexts, beliefs, local perceptions and practices related to health in the three selected communities [24,25]. The ethnographic study had two components: (1) Participant observation by the researchers living in the communities (January to April 2021) and (2) an auto-ethnographic diary study—adopted due to the disruptions caused by COVID-19 [21]—using auto-ethnographic diaries documented by community co-researchers (May to June 2021) and follow-up interviews combined with participant observations [20] (November 2021 to January 2022) (Fig 1).

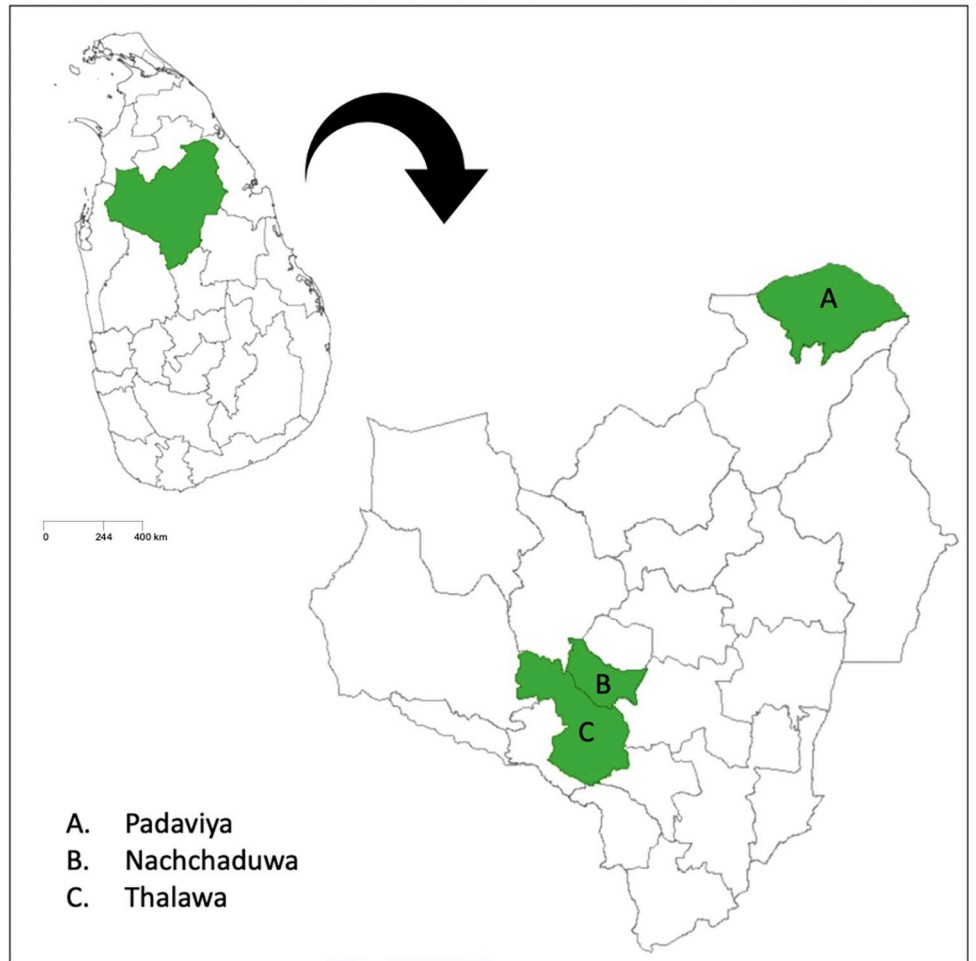

**Fig 2. The selected DS divisions within Anuradhapura district, Sri Lanka; Source of the base file: https://gadm. org/maps/LKA_1.html.**

**Participant observation.**    Three ECLIPSE researchers (including the first author, HN) were trained in ethnographic research and relocated to the field sites. The researchers identified key stakeholders in each village, conducted ethnographic interviews [26], and participated in daily communal life by engaging in everyday practices such as cooking, harvesting, observing healing and patient-healthcare worker interactions and attending religious rituals in the community. Prolonged engagement aids in building trust between the researchers and the community for a deeper understanding of community values, perceptions, communication channels, power hierarchies and ethos driving those different communities [24]. The researchers maintained daily contact with the community members through frequent visits across the communities, phone calls and attendance at all major community events (e.g., *Katina Puja*–A religious and heavily celebrated event of the offering of new robes to a Buddhist priest, *Pulleyar dhana*–An event where offerings are made for God Pulleyar after harvesting). The researchers routinely took field notes on the observations made. Through such prolonged engagement, the ECLIPSE researchers became familiar with the communities, their shared values and beliefs [24]. All these steps enhanced the quality and quantity of data collected throughout the study [27].

**The auto-ethnographic diary study with community co-researchers.**   The first major wave of the COVID-19 pandemic hit Sri Lanka in April 2021, during which the ethnographers had to leave the communities for safety reasons for both parties and ethical concerns about conducting research based on direct contact with the participants. During this period (May to November 2021), we designed and conducted an auto-ethnographic diary study with community members to continue the ethnographic study. Participant diaries are an effective way to collect data on behaviours that are inaccessible through other social research methods [28]. The use of diaries reduces the recall bias and telescoping effects that occur in retrospective interviews [29]. Combined with participant interviews, this method can be used as an approximate substitute for participant observation [30]. A similar method was previously successfully implemented in the same district [31]. The experience gained from participant observation and data analysis informed the diary design and identification of the community co-researchers.

**Participant recruitment, distribution of diaries and follow-up.**   With the established CEI, the community co-researchers were purposefully selected with key community members in field sites ensuring maximum variability in terms of socio-demographic factors, including age, gender, and occupation. They were guided to document their daily routines and observations in the diaries for one month. Community members were responsible for the distribution of the diary to the selected community co-researchers. Several community co-researchers were trained on the instructions on how to write the diary, and they were responsible for keeping up with the rest of the community co-researchers on the progress.

**Post-diary interviews.**   Based on the analysis of diaries, we prepared topic guides tailored to each person to interview the community co-researchers to clarify and elaborate on the findings documented in the diaries during the follow-up diary interviews [28,32]. The post-diary interviews were conducted from June to August 2021, when Sri Lanka temporarily lifted the COVID-19 lockdowns (Fig 1).

## Component 2: The study on people with CL

Studying the lived experiences of people with CL was a challenge during the pandemic. We developed a novel approach, Participant Experience Reflection Journal (PERJ), to capture the ethnographic details about the CL patient journey and the experiences of living with CL in rural Sri Lanka. The journals were coupled with follow-up interviews to understand the disease burden further. The community members were involved in each step of the study, including the conceptualisation.

**Participant experience reflection journal (PERJ).**   The PERJ was co-developed with the community using the CEI principles [19]. Preliminary analysis of the field notes, and auto-ethnographic diaries facilitated the development of the PERJ. The type and phrasing of questions, the layout of the journal, the questions that needed to be asked, the language, how the questions and institutions were articulated, the font style and sizes, and the space required to write an answer were discussed with several community members through several rounds of meetings and changed after CEI input. The journal consisted of open-ended questions on the psychosocial problems faced throughout the patient journey to be answered and completed written by people with CL. The journal was piloted in the community before this study component was rolled out (S1 Text).

**Participant recruitment, distribution of PERJ and follow-up.**   Facilitated by the ethnographic data, snowballing, and with the help of key community members, thirty people with CL (ten from each field site) were selected and approached to represent different age groups, gender, and different lesion size and sites (S1 Table). The participants were selected from the

same three communities where ethnography was conducted. The participants were CL patients who had completed or were undergoing treatment at a government hospital at the time of recruitment. The community members found the participants from each field site, explained the study to the participants and obtained consensus before the research team visited them. The researchers met the participants at their residences together with key community members, explained the study aims, and obtained consent. The participants who volunteered to participate were well informed of the task by the researchers. They were also provided instructions on completing the journal through an information leaflet. The participants were given a month to complete the journals (August–September 2021). The participants were followed up by weekly telephone calls and were reminded to complete the journal while clarifying any doubts they had during documentation. The key community members who were involved in designing the study provided updates about the study's progress during the study period.

**Post-PERJ interviews.**   After analysing the returned journals, individualised separate interview topic guides were developed for each participant, specifically tailored to the information written in the PERJs. The objective was to get an in-depth understanding, exploration, and elaboration of the content of each journal. The participants were then contacted to arrange a date to conduct interviews at their convenience. Interviews were conducted at participants' residences at the convenience of the participant. The length of the interview ranged from 30 mins-120 mins. The same researchers who conducted ethnographic studies in the particular field conducted the interviews for better context-specific interpretation of the interviews. We conducted the interviews once the COVID-19 restrictions were lifted from December 2021 to January 2022 (Fig 1).

## Data analysis

We analysed all four datasets (ethnographic fieldnotes, auto-ethnographic diaries, post-diary interview transcripts, PERJ, and post-PERJ interview transcripts) iteratively at different time points to supplement the ongoing study components (Fig 1). We used thematic analysis [33]. We analysed data manually in the native language itself in order to preserve and understand the exact meanings during the analysis. Two researchers first independently coded the data using five transcripts of each four types, and a coding scheme was developed. The rest of the transcripts were coded based on the coding scheme New codes were added where relevant. The autoethnographic diaries and post-diary interviews were coded using the same coding scheme and data on people with CL from PERJ and post-PERJ interviews were coded using a separate second coding scheme. When coding the field notes of participant observation, we included the culture and context-related additional codes to the first coding scheme and CL-related codes and data to the second coding scheme. After coding all the transcripts, the sub-themes and themes were generated inductively, with the consensus of both investigators. The major themes were formulated to distinguish/identify the multifaceted burden of CL in a foreseen scenario where public health interventions could be planned to minimise the different types of burden [31]. The ethnographic data were used to understand and interpret the context's psychosocial burden. A consensus was reached with the wider research group whenever necessary in a continuous manner.

## Results

### Study participants

Of the thirty community co-researchers recruited for the auto-ethnographic diary study, twenty-eight completed the diaries and follow-up diary interviews were completed for twenty-six. The participants' age range spanned from 19 to 71 years, with females aged between 29

and 71 years and males between 24 and 71 years. Among the twenty-eight participants, sixteen were female. All thirty participants with CL returned the PERJ, and twenty-five PERJ interviews were completed. Of them, twelve were females. The participants' age range varied from 18 to 75 years, with females ranging from 18 to 75 years and males ranging from 24 to 73 years. The most common site of the CL lesion was the leg. Most of the participants (21) were employed during the study. Details of study participants are in S2 Table.

## Psychosocial burden

We identified three major themes in our analysis of the psychosocial burden; The CL wound-related burden, Treatment-related burden, and Illness experience-related burden.

**The CL wound-related burden.** The participants in this study had different clinical manifestations, including small nodules to ulcerated wounds with complications (S1 Table). A 'notable wound' was of concern to many. According to the participants, a 'notable wound' could be a big wound (*loku thuwalayak*), a facial wound, multiple wounds in the body or a wound readily visible (*hondata penena thanaka thiyena thuwalayak*). In addition to their own experiences of living with CL, they also expressed their perceptions and reflections on the wounds of other people with CL whom they met in the dermatology clinic. We identified different aspects of psychosocial burden related to the wound, described in 6 sub-themes (Fig 3). (PERJ will be used to indicate that the verbatims are from the PERJ and PERJ-i will be used to indicate post-PERJ interviews)

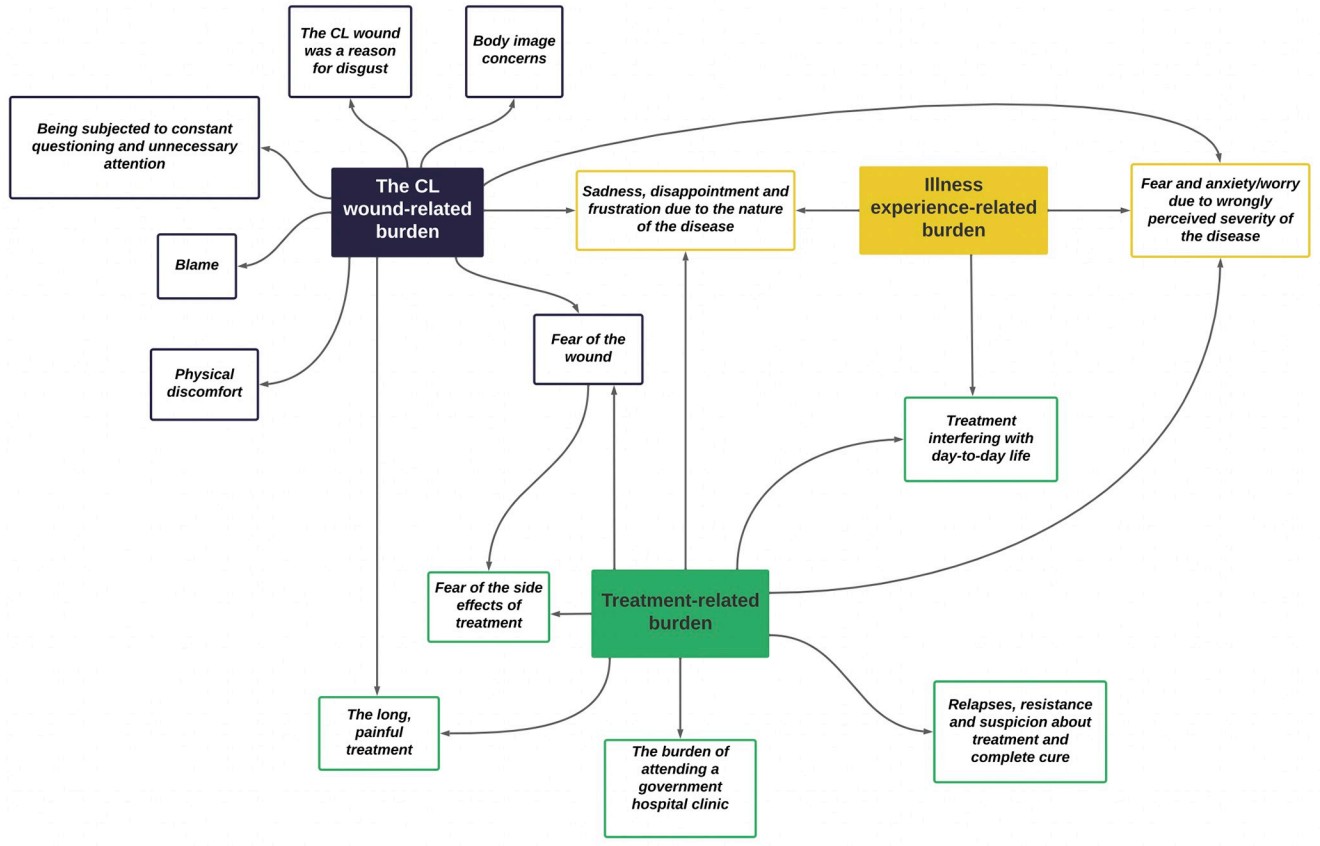

**Fig 3. The interconnectedness of the psychosocial burden of CL in rural Sri Lanka.**

**Being subjected to constant questioning and unnecessary attention.** People with 'notable wounds' faced constant questioning by others and received attention deemed 'unnecessary' and 'too much' by them. They described this as a nuisance. Some participants were tired of and disappointed in having to answer about the wound and the disease.

Few people resorted to covering up the wound to stop people from asking questions and paying attention:

> I found it hard to talk to people like I used to because of my wound. When I went to places with lots of people, I felt awkward because they might ask me about it. So, I usually tried to hide my wound. (J19/PERJ)

**The CL wound was a reason for disgust.** Some participants who had 'notable wounds' documented that they felt disgusted by their own wounds. A 44-year-old man documenting his experiences in the journal stated;

> My wound kept getting worse and started to look really bad. The wounds caused by sandfly disease can be unattractive and gross. (J19/PERJ)

People with notable wounds anticipated disgust from the community and were subjected to disgust by others as well. The site, size and colour of the wound were the reasons for disgust. It was a constant worry for people with notable lesions though they tried to express that being subjected to disgust is reasonable when you have a severe wound.

> I often feel like people will be disgusted by me because of my wound. Unfortunately, I even feel that way when I see others with wounds like mine. I feel sympathy for them because I understand what they're going through. (J10/PERJ-i)

> I find it very unpleasant to look at some people, especially those with wounds on their faces. I have seen people with wounds on their ears or black-coloured wounds on their faces, which is quite disturbing, especially if the wounds are large. (J10/PERJ-i)

None of the participants with a small nodule or papule mentioned that they were disgusted by others because they have CL.

**Body image concerns.** People explained they were distressed as they thought the wound or the scar affected their beauty.

> Every time I visit the clinic, I meet nearly a hundred people with lesions on different parts of their bodies. Once, I met a couple of young girls who were at an age where they were very concerned about their appearance. I felt sad because this disease affected their beauty, and the lesions distorted their once-beautiful faces. (J01/PERJ-i)

Distortion (*vikurthiyak*) was a word that community members commonly used to describe a wound or a person with a notable wound. This notion was attached with many consequences and burdens described above and below, such as fear, disgust, and shame. Facial wounds are seen as a distortion, and people with facial wounds are seen as ugly. People used different similes to explain the distortion; "look like bee hives, with holes and ridges, like a frog over the nose".

> At the clinic, I encountered a patient with a lesion on his nose that looked like he had a frog on it. It was a severe distortion that left me feeling dismayed and afraid. His nose was

swollen and discoloured, and I feared it might fall off. I felt sorry for him because he was a young man who must have been depressed about his situation. I'm worried about his future and pray that no one else has to go through what he is experiencing. I hope he recovers quickly. (J04/PERJ)

Participants said those with facial wounds were ashamed and tried to cover the wounds from being visible using handkerchiefs or hands.

**Blame.**   Blame was socially constructed in the communities. People with small papules/ nodules blame people with notable wounds for negligence. Furthermore, some people also mentioned that if people with notable wounds face negative reactions from society, it is their own fault as they have neglected the wound until it gets bigger and worse.

I was so shocked by some of the people with large wounds that I felt numb all over my body. I wondered how anyone could ignore a wound like that without seeking medical attention. I would have taken some medicine if it were me, even if it was just a small wound. I don't understand why they kept waiting and neglected their wound just because it wasn't causing them any pain. (J09/PERJ-i)

**Physical discomfort.**   For some people, the wound has been a painful experience. Very few people with small wounds complained of pain. However, people with chronic wounds and complications such as secondary infection and inflammation felt pain. This, coupled with the pain of the treatment (described below) process, has caused patients to bear a considerable burden. People complained of itchiness and discomfort in wearing clothes when the wound was on the leg. The presence of a wound affected the occupations of some people, such as fishermen, farmers, carpenters, and chefs. The two main reasons were that the nature of the job, such as farming or fishing, would cause the lesion to get infected or cause physical discomfort during work.

At the clinic, I talked to a person with a wound on his leg who seemed very upset about it. The wound made it impossible for him to do his daily work, and he couldn't wear trousers because they would touch the wound. As a result, he had to stop working when he had the disease. (J09/PERJ-i)

**Fear of the wound.**   Fear manifested in different ways and was a major concern among people with CL throughout. Participants who had small nodules or papules confessed that they feared other people with notable wounds for two main reasons; (i) the outlook of the wound and (ii) the thought that their lesions would also get bigger like others. People with facial wounds said they lived in fear throughout the treatment period that their wounds would get bigger.

I dreaded going to the clinic because I would encounter other patients who had frightening wounds. Some people's wounds were so big that I couldn't even look at them. It made me scared that I might develop similar wounds on my face. Going to the clinic made me feel miserable. (J07/PERJ)

**Treatment-related burden.**   The unique nature of the treatment of CL led to physical and psychological suffering and also caused many socio-economic difficulties for people with the disease. The treatment-related burden was multifaceted. We discuss the treatment-related burden here under five sub-themes.

**The long, painful treatment.** The pain from the injections was described as different and more painful than the routine injection procedures such as vaccination and venepuncture, as the drug is directly injected into the lesion, causing people with multiple/big/facial wounds to suffer more. Suffering was more pronounced when the participants observed bleeding from the wound and enlargement of the wound while the drug was injected. Especially injections to sensitive places like the neck, face, and buttocks were mentioned as an extra burden.

> The first day of treatment was excruciatingly painful, and I vowed never to go back again, even if it cost me my life. However, I had to face the difficulties of attending the clinic and enduring the pain every week. After some treatments, I would sleep the whole day because the pain was unbearable. The doctors had to give me injections directly into the lesion, and on some days, the pain was so intense that I would faint. All of this made me feel very sad. (J24/PERJ-i)

Several people who have gone through treatment failure and disease relapses mentioned that they thought about ceasing the treatment because of the painful procedure.

**Fear of the side effects of treatment.** Some believe that the treatment is 'harsh' ('*Sera behethak*'), which can cause unknown side effects. They did not have enough knowledge nor were given any information on possible side effects of the treatment, which has led to many unfounded speculations such as treatment being tocix and the belief that the lesions could worsen with the treatment before it gets healed.

> I think the medicine they use for treatment is really strong. It makes me wonder what happens when they put those shots in our bodies. (J17/PERJ-i)

**The burden of attending a government hospital clinic.** People with CL detested attending the government hospital clinic. The clinics are often crowded with 100–300 patients. Participants use the phrases; 'gruelling task', 'never felt happy', and 'was distressed to think of it' to describe the burden of attending a clinic. Many reasons underpinned their response, including the distance, transport costs, long queues and unfriendly atmosphere at some clinics, making them disappointed, helpless, angry, and sad.

> Going to the clinic was never a happy experience for me. Everyone there is sick, and the environment is negative. I never heard anything positive there. People in the queue even pushed each other sometimes. It's not how a hospital should be. It's supposed to be a place of healing, but in reality, there's no happiness there. (J04/PERJ-i)

The ethnographic study revealed the unavailability, difficulty and transport costs that the people in the three communities had to bear to reach the city. One of the main reasons for the difficulties was the nonavailability of treatment in the peripheral hospitals. Until recently, the only government hospital clinic with the treatment in the area was the Teaching Hospital in Anuradhapura city.

> Going to the clinic in Padaviya was more convenient for me. It was the closest city to where I lived, so I could go to the clinic in the morning and return home early. However, going to Anuradhapura for treatment was very challenging. I needed someone to accompany me, which was expensive, but I had to do it to get better. (J27/PERJ-i)

Participants explained that they were subjected to discrimination by healthcare workers. Some healthcare workers have shouted at people to control them. They perceived being neglected. A participant told us about an incident where he felt ashamed due to the way health-care workers addressed them and how it led to ridicule and mockery by others.

The person who gave out medicine at the clinic made us feel really bad. She called us 'hama. . .hama' (a Sinhala lay word for skin). It made us feel really low and ashamed. Even the other people waiting in line looked at us and laughed. It was really hurtful to be treated that way, especially since we already felt bad about our skin problems. (J04/PERJ-i)

**Relapses, resistance and suspicion about treatment and complete cure.** The relapses and resistance to treatment, coupled with already existing misconceptions about the disease and the treatment, have caused some people to be suspicious and worried about the treatment outcomes. Some people also believe that doctors are still experimenting with the disease. The reasons for that are the difference in treatment effectiveness from one person to the other and doctors not revealing information about the disease.

The doctors treating us for this disease don't seem to know much about it. They don't give us much information about what to expect with treatment. Normally, doctors will explain what will happen during treatment if you have another disease or a wound. But for this disease, they are still figuring things out. This made me feel worried and unsure. (J04/PERJ-i)

There were a few instances where the treatment has not worked, and they talked about feeling stressed due to the long-term treatment that never cured the disease.

**Treatment interfering with day-to-day life.** The participants often talked about how the treatment process affects their daily life economically and socially. Taking a leave every week or losing a day's worth of money for daily wage workers was mentioned as an economic burden.

In my case, it costs me 500 rupees every time I have to go to the clinic. I am a father of three children and I work as a daily wage labourer. So, when I got this disease, I had to go to the clinic and wait in long lines. This meant that I couldn't go to work and earn my daily wage. On top of that, I had to pay for transportation costs to get to the clinic. (J19/PERJ-i).

## Illness experience-related burden

Being a person with CL, the myths and beliefs about the disease and its propagation gave rise to fear, sadness, disappointment, and guilt among the participants.

**Fear and anxiety/worry due to wrongly perceived severity of the disease.** Many participants identified CL as a dangerous/deadly disease. Commonly believed misconceptions about the parasite are one of the reasons that have solidified this notion. These misconceptions have led people to believe that the disease affects their internal organs such as kidneys, liver, lungs, etc., and in extreme cases, can make you insane or that it is fatal.

My wife was worried about me when she learned that the sandfly that causes this disease lays eggs on the body. We both felt scared because we heard that if left untreated, it can damage internal organs like the kidneys, lungs, and liver and even cause death. We heard that this disease could cause all the body functions to fail, and affecting the brain and even leading to brain death. We also heard that the insect from the sandfly's eggs could travel through the spinal cord to the brain and feed on blood to survive. (J21/PERJ-i)

The interviews revealed that healthcare professionals have contributed to the notion of CL being dangerous and deadly, advising the patients to adhere to treatment. This use of negative reinforcement to promote treatment adherence has solidified the notion that it is a dangerous/deadly disease among people with CL. Some people said that they were mentally distressed after getting to know that it is deadly.

> The doctors warned me that the sandfly's disease (CL) is a serious illness that can harm my organs. When I talked to my children about how worried I was, they tried to comfort me by saying that everything would be okay since I received treatment. However, I still feel anxious because of what the doctors told me about the possible harm to my organs. (J26/PERJ-i)

The perception of the disease being dangerous was also associated with the false belief about the 'poison of the sandfly' ('*Weli massage wisa*'). Participants believed that 'the poison of the sandfly' remains on your body and make them weak. They also connect the fact that the wound takes a long time to heal to this 'poison' and believe the lesion develops from a papule to a big wound because of this poison inserted by the sandfly.

> Some people believe that if your sore or wound is not healing, it's because of the sandfly's poison. They think that the poison stays in your body and prevents the wound from getting better. (J14/PERJ-i)

**Sadness, disappointment and frustration due to the nature of the disease.** People with CL documented that it is frustrating to live with the disease. Various reasons, such as having to face negative reactions from society, the effect on day-to-day life, and the loss of work, were among the reasons. For some people, the treatment has not worked for a prolonged time, and they said they pitied themselves, and others also felt sorry for them.

> This disease made it difficult for me to do my regular farming work and other daily tasks. It was frustrating because it prevented me from being as productive as I wanted to be. Normally, people my age don't have to go to the clinic every week, and I never had to go to one before I got this disease. Having to go to the clinic every week made me feel like a sick person. (J01/PERJ)

Stigma and marginalisation were not directly described by the people but appeared when questions were asked in an indirect manner on requesting to describe the experiences of other people with CL.

> I met a man at the clinic. He has to travel a long distance to come to the clinic. He used to talk to us about his struggles. He said the villagers of his village fear this disease and that he feels like they are marginalising him. He said he tries his best to educate the villagers about this disease. (J06/PERJ-i)

The different aspects of the psychosocial burden of CL were interconnected and complex. This interconnectedness is shown in Fig 3. The complex nature of the burden caused by this disease was noticed in observations made in the ethnographic study. People were quick to point out how much of a burden this disease was due to a combination of many issues they have faced. Each CL patient faces a unique mixture of burdens according to their circumstances. The below field note excerpts indicate the multifaceted burden a female CL patient who had gone through a long treatment process,

(. . .) During our conversation, the lady who was a patient with CL repeatedly expressed the burden she felt and how her appearance was affected by both the disease and treatments. She pointed out the black spots on her body, especially on her lips, that never went away despite being treated by doctors. She showed me a 1–2 mm wide black spot on her lower lip, which appeared like a birthmark. She sarcastically and slightly angrily commented that the doctors had promised it would disappear, but it hadn't. She was mentally distressed and had no energy to work. She expressed that the treatments were harsh and made her feel as if her own body was a distortion. She didn't do any work and spent most of her time in bed. Other people she knew with the disease had also experienced difficulties in their daily lives, not because the disease was severe or the lesions were painful, but because the treatment was harsh. However, she couldn't stop taking the medication as the doctors had advised her to take it on time, no matter how painful it was. She said she even cried when the injection was given to her.

-Fieldnotes HN (November 2021)-

In identifying the patterns of CL burden, we observed that CL exerts an extra burden on extreme age groups. Many participants mentioned that it is unbearable to tolerate the pain and suffering of children. Attending regular clinics, travelling, standing in queues and bearing the pain was difficult for the elders with the disease.

To go to the clinic, I leave home very early, usually around 5:30–6:00 in the morning. I often miss breakfast because I need to catch an early bus, and sometimes I have to skip breakfast or buy something from the hospital canteen. When I arrived at the clinic, long queues were formed, and it was hard for me to stand in line due to my age. That's why I always bring my daughter or grandson with me to the clinic. There are so many people there, and it can be overwhelming. (J13/PERJ)

## Discussion

The patients' experiences reported in this study, informed by a broader CEI-based approach, reveal the complex psychosocial burden resulting from CL in resource-limited settings. Previous research done in Suriname [34,35], Iran [36], Tunisia [37], Pakistan [38]' and Columbia [39] has shown that patients with CL often face a range of negative emotions, including fear [35–37], anxiety [10,36,40], and distress [34,39], which can impact their quality of life [10,37,41–43]. The patients in this study reported how the negative emotions are related to the physical symptoms of the disease, the treatment process and the nature of the disease. The findings related to these three aspects of manifestation of burden denote direct public health links to improve people-centered care in CL.

The disease impacted the patients' daily lives, with significant disruptions to their routines, including the need for regular trips to the clinic and the difficulty of managing daily tasks due to the physical limitations caused by the disease. The ethnographic component of this study revealed the difficulties faced by people in these communities due to poor public transport services, distance to the treatment centres and impediments to their livelihoods. Such disruptions can profoundly impact the patients' sense of control and autonomy [34,37], leading to feelings of hopelessness and helplessness [44]. Farming is a common activity in these rural communities. They did not see minor cuts, wounds and insect bites as a burden because it is a part of their daily routine. As farming is a major exposure to the disease [45,46], and a notable wound causes extra burden, it is essential that people in these rural communities are empowered to

seek health at early stages, when the lesion is small, in order to disrupt the vicious cycle of psychosocial burden due to CL.

The societal and cultural norms, myths, and misinformation surrounding the disease further compound the psychosocial burden of leishmaniasis. CL is well-known to have such myths and disruptive norms [13,47]. Similar to other countries, the CL wound in Sri Lanka is considered a distortion, affecting beauty, resulting in people feeling ashamed and resorting to measures like hiding the wound [10,48,49]. Disgust is another factor that could be a potential driver of both self and social stigma [50], as observed in this study as well as other studies [47,48,51,52].

To address the psychosocial burden of leishmaniasis, a comprehensive approach, focussing not only on the physical symptoms of the disease but also the psychological and social aspects of the illness, is necessary. The biopsychosocial model of health [53–55] provides a theoretical framework for understanding the interplay between biological, psychological, and social factors in health and illness. Such an approach would require providing access to health services, promoting community awareness and education about the disease, and addressing the underlying social determinants of health that contribute to the marginalisation and stigmatisation of affected individuals [56]. We can draw lessons from success stories like the management of leprosy stigma in Sri Lanka [57,58], shifting away from solely knowledge-based approaches [59] and focusing on holistic strategies that promote attitudinal and behavioural changes. Our data shows that healthcare services for CL should be improved considering the biopsychosocial model of health and using the CEI approach will facilitate the development of interventions with a multisectoral collaborative approach assimilated within a more flexible and patient-sensitive health care service.

We propose the people-centred care approach [56] to address the psychosocial burden of leishmaniasis we observed in this study. This approach recognises the importance of treating patients as individuals with unique needs, preferences, and circumstances [60]. By engaging patients in their care, healthcare providers can better understand the personal and social factors that impact their health and well-being [61]. As an example we observe that some individuals may experience fear, shame and disgust than others which is mainly due to the lack of awareness and knowledge on the disease. The capability and intention of a treating physician to recognize and console an individual providing information about the disease could help alleviate the negative emotions and encounters. People-centred care also involves prioritising the patients' lived experiences and preferences (as we show in our results) rather than focusing solely on disease management. Systematically disseminating the findings by healthcare personnel would be the first step in driving the health workforce towards a patient-centred approach. However, extensive discussions involving different governing authorities and stakeholders are necessary to universally facilitate this, along with policy changes concerning the increased workload and time constraints in the government hospitals. Nevertheless, this will help improve patient outcomes, promote patient empowerment, and address leishmaniasis's complex psychosocial burden.

Our study is not without its limitations. Since we included people who have completed treatment already, there could be recall bias, potentially resulting in the under-reporting of more nuanced burdens that might have been forgotten over time. However, we can assume that participants might remember and document the most important when it comes to burden. Providing participants with the chance to document their experiences in a free and open environment during the PERJ and subsequently reflecting on their documentation during post-PERJ interviews would likely have reduced the likelihood of underreporting by the participants. By conducting ethnographic research and increasing familiarity, we have attempted to reduce the influence of social desirability bias [62].

Although we do not intend to generalise the findings we present here, there could be differences in the psychosocial burden of CL in an urban setting or in settings with different cultural contexts. As a study conducted in a rural area, this shows some burdens uniquely affecting rural communities. Further studies need to be done to understand the burden in urban settings.

## Conclusion

In conclusion, the psychosocial burden of leishmaniasis is a significant public health concern that requires a comprehensive approach addressing the illness's physical, psychological, and social aspects. By doing so, we can improve patient outcomes, promote equity and justice in healthcare, and ultimately contribute to a healthier society. We propose a people-centred healthcare model to mitigate the psychosocial burden of CL in these settings.

## Supporting information

**S1 Text. PERJ.**
(PDF)

**S1 Table. Socio-demographic and CL lesion characteristics of PERJ participants.**
(DOCX)

**S2 Table. Details of the auto-ethnographic diary study participants.**
(DOCX)

## Acknowledgments

The authors would like to thank the NIHR for supporting ECLIPSE. The authors would also like to acknowledge the three communities in three selected study sites for their support of research activities. Furthermore, we would like to recognise our research assistants, Mr Sandaru Hasaranga Shanthapriya and Mrs Lakshani Ranathunga, and the ECLIPSE team for their assistance given throughout the project.

## Author Contributions

**Conceptualization:** Hasara Nuwangi, Kosala Gayan Weerakoon, Suneth Buddhika Agampodi, Thilini Chanchala Agampodi.

**Data curation:** Hasara Nuwangi.

**Formal analysis:** Hasara Nuwangi.

**Funding acquisition:** Lisa Dikomitis, Suneth Buddhika Agampodi.

**Methodology:** Hasara Nuwangi, Lisa Dikomitis, Kosala Gayan Weerakoon, Suneth Buddhika Agampodi, Thilini Chanchala Agampodi.

**Project administration:** Suneth Buddhika Agampodi.

**Supervision:** Lisa Dikomitis, Kosala Gayan Weerakoon, Suneth Buddhika Agampodi, Thilini Chanchala Agampodi.

**Writing – original draft:** Hasara Nuwangi.

**Writing – review & editing:** Hasara Nuwangi, Lisa Dikomitis, Kosala Gayan Weerakoon, Suneth Buddhika Agampodi, Thilini Chanchala Agampodi.

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
