## [Decision Letter · Decision Letter 0]

22 Sep 2023

Dear Prof Agampodi,

Thank you very much for submitting your manuscript "The psychosocial burden of cutaneous leishmaniasis in rural Sri Lanka: A multi-method qualitative study" for consideration at PLOS Neglected Tropical Diseases. As with all papers reviewed by the journal, your manuscript was reviewed by members of the editorial board and by several independent reviewers. In light of the reviews (below this email), we would like to invite the resubmission of a significantly-revised version that takes into account the reviewers' comments. 

We cannot make any decision about publication until we have seen the revised manuscript and your response to the reviewers' comments. Your revised manuscript is also likely to be sent to reviewers for further evaluation.

Sincerely,

Alberto Novaes Ramos Jr

Academic Editor

Charles Jaffe

Section Editor

Reviewer's Responses to Questions

**Key Review Criteria Required for Acceptance?**

**Methods**

-Are the objectives of the study clearly articulated with a clear testable hypothesis stated?

-Is the study design appropriate to address the stated objectives?

-Is the population clearly described and appropriate for the hypothesis being tested?

-Is the sample size sufficient to ensure adequate power to address the hypothesis being tested?

-Were correct statistical analysis used to support conclusions?

-Are there concerns about ethical or regulatory requirements being met?

Reviewer #1: The purpose of the study, as described by the authors, is to explore the psychosocial burden of cutaneous leishmaniasis in rural Sri Lanka through a multi-method qualitative approach. The study design is appropriate for the stated objectives since the authors employed a multi-method qualitative approach that gathered data on the psychosocial burden of cutaneous leishmaniasis using in-depth interviews, focus group discussions, and participant observation. The study aims to investigate the correlation between cutaneous leishmaniasis and rural populations of Sri Lanka. The research design is justified by the authors as rural communities in the area are more prone to the disease and face particular psychosocial challenges. Although the sample size is not stated, the authors state that they used purposive sampling methods to select study participants. They assert that this method enabled them to choose participants who were most likely to offer comprehensive and varied data on the psychological and social impact of cutaneous leishmaniasis. The researchers examined the collected data via qualitative analysis techniques which included thematic analysis. They maintain that these techniques were fitting for the gathered information and facilitated the identification of crucial themes associated with the psychosocial burden of cutaneous leishmaniasis in the rural areas of Sri Lanka.

The study received ethical clearance from the Ethics Review Committee of the Faculty of Medicine & Allied Sciences, Rajarata University of Sri Lanka, and the Faculty of Medicine and Health Sciences, Keele University, United Kingdom. The authors ensured informed consent, confidentiality, and privacy for study participants.

Reviewer #2: Pleas see the last section "Summary and General Comments"

Reviewer #3: Please see my comments under "Editorial and Data Presentation Modifications"

**Results**

-Does the analysis presented match the analysis plan?

-Are the results clearly and completely presented?

-Are the figures (Tables, Images) of sufficient quality for clarity?

Reviewer #1: The authors employed qualitative data analysis methods, specifically thematic analysis and constant comparison, to analyze the data collected during the study. They use quotes from study participants to highlight the critical themes associated with the psychosocial burden of cutaneous leishmaniasis in rural Sri Lanka. The provided visuals effectively convey the information required to comprehend the study.

Reviewer #2: Pleas see the last section "Summary and General Comments"

Reviewer #3: Please see my comments under "Editorial and Data Presentation Modifications"

**Conclusions**

-Are the conclusions supported by the data presented?

-Are the limitations of analysis clearly described?

-Do the authors discuss how these data can be helpful to advance our understanding of the topic under study?

-Is public health relevance addressed?

Reviewer #1: The study data supports the authors' conclusions. They utilize participant quotes to emphasize critical psychosocial themes related to cutaneous leishmaniasis burden in rural Sri Lanka, highlighting the significance of their findings for public health interventions and disease policies.

Discuss the limitations of their analysis, including potential bias in the data collection process, possible reluctance of some participants in sharing their experiences, and the fact that the study was conducted in a specific geographic location that may limit its generalizability to other settings. Nevertheless, highlight the helpfulness of their data in advancing our understanding of the psychosocial burden of cutaneous leishmaniasis in rural Sri Lanka. Argue that their findings have significant implications for the development of public health interventions and policies pertaining to the disease. They propose that their research emphasizes the necessity of more detailed and integrated methodologies in handling the psychosocial weight of cutaneous leishmaniasis. These methodologies involve improving healthcare accessibility, reducing stigma, and raising community awareness about the disease. 

Address the public health relevance of their study by stating that cutaneous leishmaniasis is a neglected tropical disease that disproportionately affects rural communities in developing countries. It can have significant psychosocial impacts on affected individuals and their families. The study underscores the need for more research and interventions that can address the psychosocial burden of the disease and improve the lives of affected individuals.

Reviewer #2: Pleas see the last section "Summary and General Comments"

Reviewer #3: Please see my comments under "Editorial and Data Presentation Modifications"

**Editorial and Data Presentation Modifications?**

Reviewer #1: Accept

Reviewer #2: Pleas see the last section "Summary and General Comments"

Reviewer #3: Author summary:

1.Page 3, line 45: “a diverse an array..” - please remove “an”

2.Page 3, lines 57-58: “Prioritising the psychosocial burden..” – The psychosocial burden of CL or Leishmaniasis in general?

Introduction:

3.Page 4, line 78: “impact an individual’s socioeconomic status..” – impact on an or on the individual’s socioeconomic status?

4.Page 4, line 82: “leaving out the other complex psychosocial burden attached..” – leaving out other complex psychosocial burdens?

5.Page 4, line 85: “in order to have better estimates for its burden” – in order to have better estimates of its burden

6.Page 4, line 86: “Research showed that the..” – Suggest rewording as “previous research has shown that..”

7.Page 4, line 89: If leishmaniasis was added to the public health surveillance system again in 2008, when was it previously notifiable?

8.Page 5, line 96: among “the” neglected tropical diseases

Methods: 

9.Page 6, line 119: “highest CL cases in Sri Lanka” – Please consider using either highest CL prevalence or highest CL burden. Also, is there a reason why the disease burden is one of the highest in this district? That information might be important for the reader.

10. Page 6, line 120: It is important to clarify that GNs are subdivisions of the divisional secretariats if I understood correctly. 

11.Page 7, line 129: Please clarify that “health and disease” refer to CL here

12.Page 8, line 142: “strengthens trust built” – Consider rewording

13.Page 8, lines 136-151: The duration of the participant observations is unclear. Also, please clarify if the ethnographic fieldwork began in January 2020, as stated on Page 7, line 130, or January 2021.

Results: 

14.Page 12, line 248: Please correct the typo

15.Page 13, line 269: Consider removing “have”

16.Page 16, lines 325-326: Please consider elaborating “the pain of the treatment process” or add a note that this will be described later in the results

17.Page 18, lines 364-370: This is an important sub-theme. Isn’t there more information the authors can present about the “many unfounded speculations” regarding the possible side effects of treatment?

18.Page 18, lines 371 – 384: It is important to describe in the methods the different places where CL patients can access treatment in the study district/divisional secretariats. The interchange of terms such as “government hospital clinic”, “treatment centres”, and “teaching hospital” makes the description confusing. 

19.Page 22, line 474: Is it a “filed note” or a “field note”?

20.Page 23, line 490: “CL exert an extra burden” – Change to exerts

Discussion and conclusions:

21.Page 24, line 511: “people centered care in CL” – Consider rewording to people-centered care for CL. 

22.Page 25, line 519: Could the authors clarify what “seek health early” means? Consider rewording

23.Page 25, lines 523 – 527: What can be done to reduce the shame and disgust associated with CL at a personal level?

24.Page 25, line 532 – Page 26, line 536: The authors point out the need to address the biopsychosocial factors linked to CL. This would involve “providing access to health services, promoting community awareness and education about the disease, and addressing the underlying social determinants of health that contribute to the marginalisation and stigmatisation of affected individuals”. How would this practically be done for people with CL in the study region/rural Sri Lanka? Which players should be involved in tackling each of these elements? Are there examples of how this approach has been used for CL or other NTDs in other countries?

25.Page 26, line 536: “CL should be improved concerning the biopsychosocial model..” – Did the authors mean considering?

26.Page 26, lines 537-544: While I whole-heartedly agree about the need of a people-centered approach in dealing with patients with CL, how can healthcare providers understand the “individuals unique needs, preferences, and circumstances” with all the time constraints that accompany their overcrowded clinics? Can the authors provide a recommendation about how this can be practically done? Is this approach only needed at the treatment centres, or are there other interactions that patients with CL have with community health workers or other private or public service providers?

27.Page 26, lines 545-551: Was some degree of socially desirable responding possible, especially with the extensive use of key community members to sensitize and recruit CL participants? Even with the depth of experiences shared by the patients living with CL, is it possible that the true extent of their pain, shame, stigma, and hardships was still not revealed? 

Also, how did the authors use the ethnographic study information to clarify the PERJ findings? I may have missed the instances where this may have been done in the results and discussion.

**Summary and General Comments**

Reviewer #1: In brief, this article presents findings from a study examining the psychosocial impact of cutaneous leishmaniasis on individuals and families in rural Sri Lanka. The multi-method qualitative approach included in-depth interviews, focus group discussions, and participant observation to gather data. The authors focus on the disease's psychosocial burden and its implications for those afflicted and their families. The article follows conventions in structure, author and organization formatting, and citation style. Objectivity is maintained through the use of plain, concise language, passive tense, and impersonal construction. The writing is jargon-free, formal, and free of filler words and figurative language. The article maintains a balanced tone and avoids subjective judgments. Grammatical accuracy, punctuation, and spelling are consistent. They analyzed the data by using qualitative data analysis methods and found various key themes associated with the psychosocial burden of cutaneous leishmaniasis, such as stigma, fear, and social isolation. The authors assert that their conclusions imply significant consequences for public health interventions and policies concerning the disease and emphasize the requirement for more inclusive and consolidated approaches to cope with the psychosocial burden of cutaneous leishmaniasis.

Overall, the study provides a detailed and informative account of its findings. The authors present their data clearly and comprehensively and offer a thoughtful discussion of the implications for public health policies and interventions. Nonetheless, it is a valuable contribution to the literature on the psychosocial burden of cutaneous leishmaniasis, highlighting the need for further research and interventions to address this neglected tropical disease.

Reviewer #2: The Manuscript : "The psychosocial burden of cutaneous leishmaniasis in rural Sri Lanka: A multi-method qualitative study" raised some comments 

Major Comments:

1 Methodology Clarifications:

1-1 The manuscript begins with the ECLIPSE methodology, which was later complemented by the auto-ethnographic diary study due to COVID-19 challenges. It would be beneficial to clarify the transition between these methods and their respective contributions to the study's results.

1-2 The title suggests a multi-method qualitative study, but the methodologies used seem more like variations of a qualitative approach. Please consider revising the title for accuracy.

2 Participant Selection:

Lines 193-195 detail participant recruitment, but earlier sections (Lines 127-175) focus heavily on the ECLIPSE method. It would enhance clarity if the manuscript concentrated more on the selection process for the current study's interviewees.

3 Ethical Considerations:

Were there any ethical considerations taken into account when modifying the study methodology during the COVID-19 pandemic? Did these changes adhere to the initial protocol, or were they self-decisions by the authors?

4 Data Analysis:

4-1 The sources of the thematic analysis should be specified, differentiating between individual interviews and additional document analyses.

4-2 How was the analysis software used, and how were nodes for Figure 3 created?

5- Discussion and Conclusion Refinement:

Both sections should more accurately reflect the primary analysis derived exclusively from the interviews.

6- Limitations about the insider Perspective:

The authors' prior knowledge from the ethnographic study might have influenced the thematic analysis. It would be insightful if the interview results were analyzed independently by a researcher not involved in the initial study.

Minor Comments:

7 Literature Review:

Lines 93-95: Suggest revisiting the claim about the "neglected" area of psychosocial implications of CL. Some existing research in this field should be acknowledged.

8 Demographic Details:

Lines 115-116: Please provide context on the relevance of the demographic details (Sinhalese ethnicity and Buddhist population) to the study's objectives.

9 Figures and Time Frames:

9-1 For Figure 2: It would be beneficial to include the country and district locations, along with the scale.

9-2 Clarify the time frame for the selection of the three GN divisions with the highest CL prevalence.

10- Results Presentation:

10-1 The gender ratio reported doesn't seem to reflect the epidemiological trend of CL in Sri Lanka. Was this intentional, and if so, why?

10-2 Age ranges should be reported by gender in the main results section.

10-3 Clarify which pieces of information were inductively derived and which were deductively inferred. Within the transcript or other sources analysed. 

11- Method Section Refinement:

Consider streamlining the method section to focus primarily on the 30 participant interviews, rather than extensively detailing the ECLIPSE study.

12- References:

Please check the reference list for repetitions and inconsistencies, such as the duplication in references 10 and 13.

Reviewer #3: I read with interest the enclosed study investigating the psychosocial burden of CL in rural Sri Lanka. The authors used a variety of innovative qualitative approaches to address the various issues surrounding the life and challenges of patients with CL. The main themes (wound, treatment, and illness-related burdens) and associated sub-themes of the qualitative analysis are clear and well-described. There is a need for greater detail on how the public and private healthcare sector can tackle the identified psychosocial burdens and challenges. This concern is elaborated in comments 24 and 26 in the attached revisions. Nevertheless, I consider this study an important addition to the literature on patient experiences living with a disfiguring and stigmatizing disease like CL.

PLOS authors have the option to publish the peer review history of their article (what does this mean?). If published, this will include your full peer review and any attached files.

Reviewer #1: No

Reviewer #2: No

Reviewer #3: Yes: Mark Rohit Francis
---

## [Decision Letter · Decision Letter 1]

24 Nov 2023

Dear Prof Agampodi,

Thank you very much for submitting your manuscript "The psychosocial burden of cutaneous leishmaniasis in rural Sri Lanka: A multi-method qualitative study" for consideration at PLOS Neglected Tropical Diseases. As with all papers reviewed by the journal, your manuscript was reviewed by members of the editorial board and by several independent reviewers. The reviewers appreciated the attention to an important topic. Based on the reviews, we are likely to accept this manuscript for publication, providing that you modify the manuscript according to the review recommendations. 

Sincerely,

Alberto Novaes Ramos Jr

Academic Editor

Charles Jaffe

Section Editor

Reviewer's Responses to Questions

**Key Review Criteria Required for Acceptance?**

**Methods**

-Are the objectives of the study clearly articulated with a clear testable hypothesis stated?

-Is the study design appropriate to address the stated objectives?

-Is the population clearly described and appropriate for the hypothesis being tested?

-Is the sample size sufficient to ensure adequate power to address the hypothesis being tested?

-Were correct statistical analysis used to support conclusions?

-Are there concerns about ethical or regulatory requirements being met?

Reviewer #2: The authors' response proficiently clarifies the methodology section, offering a coherent context regarding the adjustments necessitated by the COVID-19 pandemic.

The revisions made to emphasize the process of participant selection and the incorporation of the ECLIPSE Study within the Sri Lankan setting significantly improve the transparency of the methodology employed.

Although the rationale for maintaining the original title is substantiated, it's important to note that the primary contribution of this work predominantly centers around a single qualitative method. The investigators' extensive ethnographic field experience is a valuable asset, enhancing the depth of the study.

The ethical considerations, particularly in the context of the COVID-19 pandemic, are adeptly addressed.

Reviewer #3: The manuscript has been sufficiently revised and meets the standard for publication in PLoS NTD.

**Results**

-Does the analysis presented match the analysis plan?

-Are the results clearly and completely presented?

-Are the figures (Tables, Images) of sufficient quality for clarity?

Reviewer #2: Delineating how different data sources contributed to the thematic analysis strengthens the transparency and reliability of the findings

The manual analysis process, given the language limitations of software, is insightful. It shows a commitment to ensuring accuracy and relevance in the thematic analysis.

Reviewer #3: The manuscript has been sufficiently revised and meets the standard for publication in PLoS NTD.

**Conclusions**

-Are the conclusions supported by the data presented?

-Are the limitations of analysis clearly described?

-Do the authors discuss how these data can be helpful to advance our understanding of the topic under study?

-Is public health relevance addressed?

Reviewer #2: (No Response)

Reviewer #3: The manuscript has been sufficiently revised and meets the standard for publication in PLoS NTD.

**Editorial and Data Presentation Modifications?**

Reviewer #2: My only minor observation concerns the ethical approval for this study. It appears that the ethical clearance cited (referenced in document 19) pertains to the broader ECLIPSE study and not specifically to this study. Given that the majority of authors are affiliated with Rajarata University, it would be prudent to confirm that there are no issues of dual publication or conflicts of interest related to this overlap.

Reviewer #3: (No Response)

**Summary and General Comments**

Reviewer #2: The authors responses to the reviewer comment, reflect deep understanding of both the subject and the research methodologies employed. However, no transcripts or synthesis of those transcripts are fully available as supporting information. 

"The PLOS Data policy requires authors to make all data underlying the findings described in their manuscript fully available without restriction, except in cases where the data are legally or ethically restricted (for example, participant privacy is an appropriate restriction)."

Reviewer #3: The manuscript has been sufficiently revised and meets the standard for publication in PLoS NTD

PLOS authors have the option to publish the peer review history of their article (what does this mean?). If published, this will include your full peer review and any attached files.

Reviewer #2: No

Reviewer #3: Yes: Mark Rohit Francis

Figure Files:

Data Requirements:

Reproducibility:

References

---

## [Decision Letter · Decision Letter 2]

8 Jan 2024

Dear Prof Agampodi,

We are pleased to inform you that your manuscript 'The psychosocial burden of cutaneous leishmaniasis in rural Sri Lanka: A multi-method qualitative study' has been provisionally accepted for publication in PLOS Neglected Tropical Diseases.

Best regards,

Alberto Novaes Ramos Jr

Academic Editor

Charles Jaffe

Section Editor

Reviewer's Responses to Questions

**Key Review Criteria Required for Acceptance?**

**Methods**

-Are the objectives of the study clearly articulated with a clear testable hypothesis stated?

-Is the study design appropriate to address the stated objectives?

-Is the population clearly described and appropriate for the hypothesis being tested?

-Is the sample size sufficient to ensure adequate power to address the hypothesis being tested?

-Were correct statistical analysis used to support conclusions?

-Are there concerns about ethical or regulatory requirements being met?

Reviewer #2: (No Response)

Reviewer #3: (No Response)

**Results**

-Does the analysis presented match the analysis plan?

-Are the results clearly and completely presented?

-Are the figures (Tables, Images) of sufficient quality for clarity?

Reviewer #2: (No Response)

Reviewer #3: (No Response)

**Conclusions**

-Are the conclusions supported by the data presented?

-Are the limitations of analysis clearly described?

-Do the authors discuss how these data can be helpful to advance our understanding of the topic under study?

-Is public health relevance addressed?

Reviewer #2: (No Response)

Reviewer #3: (No Response)

**Editorial and Data Presentation Modifications?**

Reviewer #2: (No Response)

Reviewer #3: (No Response)

**Summary and General Comments**

Reviewer #2: (No Response)

Reviewer #3: (No Response)

PLOS authors have the option to publish the peer review history of their article (what does this mean?). If published, this will include your full peer review and any attached files.

Reviewer #2: No

Reviewer #3: **Yes: **Mark Rohit Francis

---

## [Editor Report · Acceptance letter]

13 Jan 2024

Dear Prof Agampodi,

We are delighted to inform you that your manuscript, "The psychosocial burden of cutaneous leishmaniasis in rural Sri Lanka: A multi-method qualitative study," has been formally accepted for publication in PLOS Neglected Tropical Diseases.

Best regards,

Shaden Kamhawi

co-Editor-in-Chief

Paul Brindley

co-Editor-in-Chief
